# Changes in Hippocampus and Amygdala Volume with Hypoxic Stress Related to Cardiorespiratory Fitness under a High-Altitude Environment

**DOI:** 10.3390/brainsci12030359

**Published:** 2022-03-08

**Authors:** Zhi-Xin Wang, Rui Su, Hao Li, Peng Dang, Tong-Ao Zeng, Dong-Mei Chen, Jian-Guo Wu, De-Long Zhang, Hai-Lin Ma

**Affiliations:** 1Plateau Brain Science Research Center, Tibet University/South China Normal University, Lhasa 850012, China; wangzhixin666@outlook.com (Z.-X.W.); srsisu2011@163.com (R.S.); futanghu888@126.com (H.L.); 18932689504@163.com (P.D.); tsangtongao@gmail.com (T.-A.Z.); chendongmei0228@163.com (D.-M.C.); 2Management Department, Tibet Police College, Lhasa 850012, China; wjg198228@163.com; 3Key Laboratory of Brain, Cognition and Education Sciences, Center for Studies of Psychological Application, Guangdong Key Laboratory of Mental Health and Cognitive Science, Ministry of Education, School of Psychology, South China Normal University, Guangzhou 510631, China

**Keywords:** high altitude, cardiorespiratory fitness, stress, amygdala, hippocampus

## Abstract

The morphology of the hippocampus and amygdala can be significantly affected by a long-term hypoxia-induced inflammatory response. Cardiorespiratory fitness (CRF) has a significant effect on the neuroplasticity of the hippocampus and amygdala by countering inflammation. However, the role of CRF is still largely unclear at high altitudes. Here, we investigated brain limbic volumes in participants who had experienced long-term hypoxia exposure in Tibet (3680 m), utilizing high-resolution structural images to allow the segmentation of the hippocampus and amygdala into their constituent substructures. We recruited a total of 48 participants (48 males; aged = 20.92 ± 1.03 years) to undergo a structural 3T MRI, and the levels of maximal oxygen uptake (VO_2max_) were measured using a cardiorespiratory function test. Inflammatory biomarkers were also collected. The participants were divided into two groups according to the levels of median VO_2max_, and the analysis showed that the morphological indexes of subfields of the hippocampus and amygdala of the lower CRF group were decreased when compared with the higher CRF group. Furthermore, the multiple linear regression analysis showed that there was a higher association with inflammatory factors in the lower CRF group than that in the higher CRF group. This study suggested a significant association of CRF with hippocampus and amygdala volume, which may be related to hypoxic stress in high-altitude environments. A better CRF reduced physiological stress and a decrease in the inflammatory response was observed, which may be related to the increased oxygen transport capacity of the body.

## 1. Introduction

Under a high-altitude environment, the decrease in atmospheric pressure and the consequent drop in the partial pressure of oxygen (PO_2_) can result in hypobaric hypoxia. The human brain is the most oxygen-consuming organ, and is very susceptible to hypoxic stress [1]. Hypoxic stress has serious effects on brain structures, such as the hippocampus and amygdala [2,3,4]. Behavior and brain structure changes arising from hypoxia can be observed in a real high-altitude environment, and examined in simulated hypoxia situations. The hypoxia impact on the brain exhibits a significant altitude-dependent effect [5,6].

Activation of the hypothalamic–pituitary–adrenal axis (HPA) is a hallmark of the stress response [7]. At the level of the organism, a hypoxic challenge is perceived as a non-specific stress, and hypoxia could upregulate the setpoint of the HPA axis and augment adrenal steroidogenic production, resulting in neuroinflammation and neuronal cell death [8]. High-altitude hypoxia stress affects a wide range of brain areas [4,9]. Major brain regions with structural and functional abnormalities are particularly vulnerable to hypoxic stress, including the amygdala and hippocampus [8]. Changes in the amygdala and hippocampus under high-altitude hypoxia stress may be due to the influence of inflammation [10,11]. The inflammatory response to hypoxia leads to the death of hippocampal neurons [12,13]. Similar studies have found that after hypoxia stress, the inflammatory response of the hippocampus and amygdala increase, leading to the death of neurons [14,15]. Studies of hypoxia stress in the brain caused by human disease have also found that inflammation can trigger neuron damage in the hippocampus and amygdala [14]. The number of corticotropin-releasing factor- (CRF-) and neuropeptide-Y- (NPY-) positive neurons were found to be decreased in the amygdala after hypoxia-ischemia [16]. In conclusion, inflammation caused by altitude hypoxia stress seriously affects the hippocampus and amygdala.

Cardiorespiratory fitness (CRF) is an objective measure of habitual physical activity that reflects the overall capacity of the cardiorespiratory system, and has been used to assess the relationship between physical activity and health status [17]. Increasing evidence has shown that higher levels of CRF are related to better brain health [18]. Improving CRF can effectively maintain the axial homeostasis of the HPA and reduce the inflammatory response [19,20]. CRF reduces stress-induced inflammation and increases neuroplasticity in the hippocampus and amygdala [21]. For example, higher CRF was found to be associated with greater GM volumes in several AD-relevant brain regions, including the hippocampus and amygdala [22]. The CRF measured using VO_2max_ was reported to be associated with the volumetric enlargement of the hippocampal head, specifically the head region of CA1 [23]. Studies in adolescents and older adults have shown that CRF levels are associated with a greater hippocampal volume [24,25]. CRF protects neurons in the amygdala and hippocampus against Alzheimer’s disease-related degeneration, probably via enhancements of brain-derived neurotrophic factor (BDNF) signaling pathways and Aβ clearance [26].

Cardiorespiratory function is an effective indicator for evaluating oxygen transport capacity [27]. Cardiorespiratory function is a term that can be used interchangeably with CRF, and indicates the VO_2max_ or ability to undertake aerobic exercises [28]. Recent studies have shown that the acclimatization and adaptive processes at high altitude in healthy individuals, with tissue hypoxia, often lead to compromised arterial oxygenation [29]. However, CRF can improve oxygen transport at high altitudes [30]. This may be associated with a lower mean RBC age, thereby improving oxygen release and increasing tissue oxygen supply [31]. At the same time, the increase in CRF improves the affinity between hemoglobin and oxygen [32]. An increase in CRF is accompanied by an increase in brain blood flow and brain metabolism [33]. In summary, CRF is an important indicator of the cardiovascular system’s ability to deliver oxygen to peripheral tissues, and the tissue’s ability to use that oxygen [34]. Not surprisingly, CRF is involved in adaptation to high altitudes [35].

However, it is still largely unclear whether and how the CRF regulates the impact of hypoxia on the brain under high-altitude hypoxia environments. The present study aimed to explore the relationship between the volume of the hippocampus and amygdala with CRF across participants under a high-altitude environment in Tibet (3680 m). The study recruited participants who had been exposed to high altitudes for more than 2 years. VO_2max_ was used to measure CRF via a specialized cardiorespiratory function test system, and participants were divided into high and the low-CRF groups based on levels of median VO_2max_. MRI data were collected to segment the hippocampus and amygdala volumes, and the relationship between the hippocampal and amygdala subregions and the VO_2max_ was evaluated. To further explore the effect of CRF on the hippocampus and amygdala, we also collected biochemical indicators related to inflammation and immunity to identify the physical essence of the linkage of the hippocampus and amygdala with CRF in these immigrant participants.

## 2. Materials and Methods

### 2.1. Participants

This study recruited a total of 48 right-handed male participants who were born in and grew up in low altitude areas, and had migrated to high-altitude areas (Lhasa, 3680 m) for more than 2 years. The participants were divided into the high-CRF group (*n* = 23) and the low-CRF group (*n* = 25) according to their median VO_2max_. In addition, the sample size was reasonable, which was measured using the G*Power (*t* = 2.01, effect size was 0.83).

All the participants had normal vision or corrected vision, and none of them had a history of mental illness or major diseases such as traumatic brain injury, hypertension, or heart disease. The two groups were matched on age (20.92 ± 1.03 years) and years of education (14.13 ± 0.33 years) (Table 1).

This study was approved by the local ethics committee of Tibet University, and conducted in accordance with relevant guidelines and regulations. All the participants voluntarily participated in the experiment, signed the informed consent before the experiment, and received a payment after the end of the experiment.

### 2.2. Experimental Design

Forty-eight male participants who had lived in Lhasa for more than two years were randomly selected, and their VO_2max_ was measured using a cardiopulmonary exercise test (CPET). Participants were divided into the low- and high-CRF groups based on their median VO_2max_. MRI data and biochemical indicators related to inflammation were collected. Then, we tested the difference between the two groups on the subfield volume and the biochemical indicators. An across-subject regressive analysis between the biochemical parameters and the regions of the hippocampal and amygdala subfields was calculated within each group (Figure 1).

### 2.3. Maximum Oxygen Uptake

CRF was assessed with CPET. Oxygen, carbon dioxide, and respiratory flow data were collected in two conditions: during rest state and during exercise state (increased load pedal powered bicycle). During the test condition, an incremental protocol with a 30 W per two minutes stepwise was used to test exhaustion on a cycle ergometer. The power bike load was carried out in frequency independent mode at 60 RPM, with an accuracy of 5 W. In the preparation stage, the participants were asked to sit quietly for five minutes, and in the recovery phase, the original load was terminated and changed to 30 W. Heart rate and oxygen uptake were measured continuously during the CPET (MetaLyzer 3B, Cortex Medical GmbH, Leipzig, Germany), and the relevant indicators were calculated according to the standard Wasserman formula [36]. The VO_2max_ (mL·kg·min^–1^) values were applied to completely characterize the aerobic predispositions of the participants. Prior to the measurement of each participant, the device was recalibrated [37]. Maximal efforts were assumed when the participant felt exhausted, had a heart rate greater than 180, or had a respiratory exchange rate (RER) equal to or exceeding 1.1.

### 2.4. MRI Data Acquisition

T1 weighted images were collected by a Siemens 3 Tesla Allegra MRI scanner at the Tibet Armed Police Corps Hospital using a magnetization-prepared rapid gradient-echo (MP-RAGE) sequence with the following parameters: slice thickness = 1 mm, TR = 1900 ms, TE = 2.41 ms, FA = 9°, FOV = 256 mm, matrix = 256 × 256, slices = 192, voxel size = 1 mm^3^.

T1 weighted data were processed using Freesurfer 7.1.1 (http://surfer.nmr.mgh.harvard.edu, accessed on 25 February 2022) and MATLAB 2014 Runtime (https://surfer.nmr.mgh.harvard.edu/fswiki/MatlabRuntime, accessed on 25 February 2022). The standard volumetric pipeline was used to generate several files, including the Talairach transformation matrix for hippocampal and amygdaloid subfields segmentation. After the volumetric pipeline, quality control was performed to manually check the results of the brain extraction, Talairach transformation, and brain segmentation.

The automated segmentation of the hippocampal and amygdaloid subfields was driven by a probabilistic atlas and a Bayesian inference model, which maximized the probability of the segmentation [38,39]. A total of 64 structural subfields were extracted, including 20 amygdaloid subfields and 44 hippocampal subfields, as shown in Figure 2. The subfields of the amygdala included the mean volume of the lateral nucleus, basal nucleus, accessory basal nucleus, anterior amygdaloid area, central nucleus, medial nucleus, cortical nucleus, corticoamygdaloid transition, and paralaminar nucleus, and the whole amygdala was located at the bilateral hemispheres. The total of 44 hippocampal subfields included the mean volume of the hippocampal tail, subiculum body, cornuammonis (CA) 1 body, CA1 head, subiculum head, hippocampal fissure, presubiculum head, presubiculum body, parasubiculum, molecular layer head, molecular layer body, granule cell layers of the dentate gyrus (GC-DG) head, GC-DG body, CA3 body, CA3 head, CA4 head, CA4 body, fimbria, hippocampal amygdala transition area, hippocampal body, and hippocampal head, and the whole hippocampus was located in the bilateral hemispheres. Then, the volumes of the amygdala subfields and hippocampus subfields were extracted for statistical analysis.

### 2.5. Biochemical Indicators

The biochemical indicators related to inflammation were collected from the participants by venous blood sampling in Fukang Hospital, affiliated with Tibet University (Lhasa, Tibet), corresponding to a previous study in our lab [40].

### 2.6. Statistical Analysis

Statistical analyses of the hippocampal subfields’ volume, amygdala subfields’ volume, and biochemical indicators were performed with SPSS (SPSS 20, inc./IBM, Armonk, NY, USA). We carried out the Shapiro-Wilk test first and found that the data was normally distributed (*p* > 0.05). The hippocampal subfield volume and amygdala subfield volume were tested using the analysis of covariance (ANCOVA). The ratios of the hippocampal and amygdala volume to intracranial volume were used as covariables [41]. An independent sample T test was used for the biochemical indicators, and the alpha level was set at *p* < 0.05.

To test our hypothesis, the standard multiple linear regression analysis was used to analyze the relationship between the biochemical parameters and the volume of the hippocampus and amygdala subfields in each group, in which the biochemical parameters were employed as the independent factors and the volume of the target regions as the dependent factor. Notably, only the related biochemical parameters of the existing phase were input into the stepwise regression model, and the regression analyses were controlled for age and ratio of hippocampal and amygdala volume to intracranial volume by the addition of variables into the linear model as covariates [42]. Durbin-Watson tests were performed to ensure independence of errors (residuals). We checked the tolerances and correlation coefficients to make sure that there were no collinearity problems in our data set. The assumptions of linearity, error independence, homoscedasticity, outliers, and residual normality had to be satisfied before the results which could be interpreted. The multiple linear regression hypothesis was satisfied. The significance level was assumed at *p* < 0.05, and *p*-values were corrected for multiple comparisons using the FDR correction [43].

## 3. Results

### 3.1. Hippocampal and Amygdala Subfields

High CRF (2.18 ± 0.20) was significantly higher than that of the low CRF (1.62 ± 0.15) (*t* = −10.84, *p* < 0.001; Figure 3A). Analysis of covariance (ANCOVA) revealed that the left GC_ML_DG head volume (*F* = 4.71, *p* = 0.035), the left CA4 head volume (*F* = 4.75, *p* = 0.035), the right subiculum body volume (*F* = 4.34, *p* = 0.043), and the right presubiculum body volume (*F* = 9.06, *p* = 0.004) in the hippocampus of the high-CRF group were significantly larger than those of the low-CRF group. As regards the amygdala subfields, we found that the left corticoamygdaloid transition volumes (*F* = 0.465, *p* = 0.036) in the high-CRF group were significantly larger than in the low-CRF group (Figure 3B).

### 3.2. Biochemical Indicators

An independent sample T test revealed that the direct bilirubin (DBIL), total bilirubin (TBIL), and red blood cells (RBC) in the high-CRF group were significantly greater than those in the low-CRF group. The standard deviations in the red cell distribution width (RDW-SD) in the high-CRF group were significantly lower than those of the low-CRF group. There were no significant differences in the other indicators (Table 2).

### 3.3. Multiple Regression Analysis

#### 3.3.1. Hippocampus Subfields

Multiple linear regression analyses in the two CRF groups, adjusted for hippocampal total volume, were used to investigate possible associations between amygdala subfields volumetrics and biochemical parameters (Table 3 and Table 4). The regression analysis within the two groups showed that HGB (hemoglobin) effectively positively predicted the volumes of the right subiculum body (Beta = 0.70, r^2^ = 0.11, *p* < 0.05, FDR corrected) and right presubiculum body (Beta = 0.55, r^2^ = 0.11, *p* < 0.05, FDR corrected). In the low-CRF group, the eosinophil (EO) predicted the left GC_ML_DG head volume (Beta = −0.53, r^2^ = 0.37, *p* < 0.05, FDR corrected). The EO (Beta = −0.49, r^2^ = 0.38, *p* < 0.05, FDR corrected) predicted the left CA4 head volume. The NEUT predicted the right subiculum body (NEUT, Beta = 0.48, r^2^ = 0.23, *p* < 0.05, FDR corrected) and the right presubiculum body volume (Beta = 0.73, r^2^ = 0.53, *p* < 0.05, FDR corrected). In the high-CRF group, the direct bilirubin (DBIL) predicted the left GC_ML_DG head (Beta = −0.45, r^2^ = 0.25, *p* < 0.05, FDR corrected) volume and the left CA4 head (Beta = −0.44, r^2^ = 0.19, *p* < 0.05, FDR corrected) volume.

#### 3.3.2. Amygdala Subfields

Multiple linear regression analysis (adjusted amygdala total volume) showed that the amygdala and biochemical parameters were also related in the two CRF groups (Table 4). In the low-CRF group, the EO predicted the left corticoamygdaloid transition volume (Beta = −0.626, r^2^ = 0.21 *p* < 0.05, FDR corrected).

## 4. Discussion

To our knowledge, this is the first study to explore the effects of CRF on hippocampus and amygdala volumes under a real high-altitude environment. This study found that there were significant differences in the hippocampal and amygdala subregions related to CRF levels. When compared with the low-CRF group, the volumes of the hippocampus and amygdala subfields were improved in the high-CRF group, and these changes were associated with lower inflammatory responses. This suggests the existence of a relationship between CRF levels and volume changes in the hippocampus and amygdala in high-altitude environments, which may be associated with hypoxia stress.

Previous studies have found that hypoxia severely affects gray matter volumes in the hippocampus and amygdala [44]. In this study, we found that the gray matter volume in the hippocampal subfields and amygdala subfields were increased in the high-CRF group when high-altitude participants were divided into two groups according to median VO_2max_. This indicates that resilient cardiorespiratory function in high-altitude environments effectively protects against the negative effects of high altitude and a low oxygen environment on the hippocampus and amygdala. Previous literature has focused on analysis of the relationship between the volume of subcortical brain structures and CRF, and have found that higher levels of CRF are associated with greater volumes in the hippocampus and basal ganglia [45,46]. Correspondingly, increased CRF is associated with an increase in general cortical thickness [47], and higher CRF levels are associated with higher brain-derived neurotrophic factor (BDNF) levels, especially in the hippocampus. Increased hippocampal volume is positively correlated with BDNF levels [22]. Additionally, higher CRF is associated with increased volume of the hippocampus and amygdala in Alzheimer’s patients [22]. Better CRF improves the brain and behavior, as well as neurogenesis, in both healthy and dementia models, reduces toxicity and cerebral amyloids, and reduces inflammation and oxidative stress [48]. Consistent with these prior investigations, our findings further suggested that high CRF is significantly related to increased gray matter volume in the hippocampus and amygdala at high altitude among immigrant participants.

We found that the low-CRF group was more affected by inflammation than the high-CRF group. Eosinophil effectively predicted the volume of the amygdala and hippocampal subfields in the low-CRF group. The release of cortisol relates the secretion of cytokines, especially interleukin-5, which stimulates the production and differentiation of granulocytes, such as eosinophils [49]. The inflammatory response induced by HPA axis disorder affects the volume of the hippocampus and amygdala under the stress state [50]. Inflammation is often accompanied by the apoptosis of a large number of cells and changes in the nervous system [51]. Eosinophils are important markers of inflammation and are associated with damage to neurons in the hippocampus and amygdala [52,53]. Eosinophils were observed in the hippocampal and amygdala neuron damage induced by state epilepsy in mice [54]. Previous studies have shown that improved lung function is associated with decreased eosinophils [55]. We also found that NEUT were effective predictors of hippocampal subfields volume in the low-CRF group. NEUT, as phagocytes, play an important role in inflammatory immune regulation [56]. Tissue damage caused by inflammation leads to the excessive activation of NEUT, which leads to an aggravated inflammatory response [57]. The accumulation of neutrophils in the brain has been associated with increased secondary brain damage and poor neurological outcomes [58]. Traumatic brain injury results in an inflammatory response in the brain, accompanied by an influx of neutrophils into the cerebral cortex and especially the hippocampus [59]. A lower CRF was also related to greater WBC, as well as neutrophil, lymphocyte, and monocyte, counts [60]. Higher CRF reduces NEUT content and improves the immune response [61,62]. Other studies have found that higher CRF produces oxidative damage in neutrophils and induces antioxidant defenses in lymphocytes [63]. Our results showed that the hippocampus and amygdala in the low-CRF group were more susceptible to inflammation at high altitudes, resulting in smaller amygdala and hippocampal volumes. However, the relationship between these parameters and the amygdala and hippocampus was not found in the high-CRF group, suggesting that high CRF can reduce the inflammatory response and improve the plasticity of the hippocampus and amygdala under high-altitude stress, which is similar to the findings in previous studies [64,65].

Oxygen is transported primarily by hemoglobin in red blood cells [66,67]. We found RBC and HGB were higher, and RDW-SD smaller, in the high-CRF group. RDW is a measurement of the size variation, as well as an index of the heterogeneity, of the erythrocytes (i.e., anisocytosis). Higher RDW values reflect greater variation in RBC volumes and were found to be related to many inflammation diseases in previous studies [68]. Because the erythrocyte represents the body’s oxygen carrier, its redox and metabolic status is extremely important for the functioning and regulation of oxygen affinity to hemoglobin, which is determined by a number of metabolites within the erythrocyte. All tissues are dependent on RBC function, especially neurons, which use 20% of the total oxygen consumed [69]. Reduced hemoglobin levels in the hippocampus and other neurons have been found in studies of neurodegenerative diseases [70]. Increases in erythrocytes and hemoglobin are accompanied by increased hippocampal oxygenation under hypoxia [71]. On the other hand, RBCs mediate the immune system’s ability to reduce inflammation and stress [72]. Another study also found that the red blood cells of physically active rats were more resistant to oxidative stress after they were deprived of oxygen [73]. Our previous study found that healthy RBC at high altitude can support the immune system [40]. Our results here showed that individuals with high CRF levels had larger hippocampal and amygdala volumes than those with low CRF levels, possibly because individuals with high CRF levels have higher oxygen transport capacity and higher immune levels.

In addition, we found that the DBIL and TBIL of the high-CRF group negatively predicted the hippocampal subfield volume, and the DBIL and TBIL in the high-CRF group was significantly increased. The increase in DBIL and TBIL may be caused by the increase in RBC and HGB. Bilirubin has been commonly considered to be simply the “final product” of heme catabolism [74]. The rise in bilirubin in rats exposed to high altitudes may be due to an increase in red blood cell counts [75]. Bilirubin is an endogenous antioxidant that plays an important role in the anti-oxidative stress and anti-inflammation of neurons [76,77]. Bilirubin enhances the bactericidal ability of neutrophils [78]. There is evidence that it protects the cardiovascular system, neuronal systems, the hepatobiliary system, the pulmonary system, and the immune system [74]. Mildly elevated serum bilirubin is generally associated with the attenuation of oxidative stress and with a decreased inflammatory status [79]. On the other hand, a decrease in bilirubin during hypoxia is associated with increased inflammation [80]. The accumulation of bilirubin acts as an effective defense mechanism against stress and increased inflammation [81]. In conclusion, our results suggest that higher CRF at high altitudes can reduce the hypoxic stress response, thereby inducing immunity and reducing inflammation, which may be related to increased oxygen transport capacity of the body.

There were several issues which should be considered in future works. First, a longitudinal trace study should be included to validate the present findings, in which the behavioral data including the gender difference, life style, personality, and so on should be considered. Second, the physiological characteristics of hypoxia tolerance are closely related to the genome, and this could provide a potential way to better understand the molecular mechanisms of human adaptation to high altitudes [82]. Third, the neuroinflammatory markers and the role of epigenetics should be explored related to the effects of exercise and oxygen interventions on the high-altitude participants for altitude adaptation.

## 5. Conclusions

This study indicated that higher CRF can significantly protect against the decrease in hippocampal and amygdala volume induced by high-altitude hypoxia. A better CRF might be related to a lower physiological stress response. These findings provide new insights into the stress response to hypoxia in human adaptation.

## Figures and Tables

**Figure 1 brainsci-12-00359-f001:**
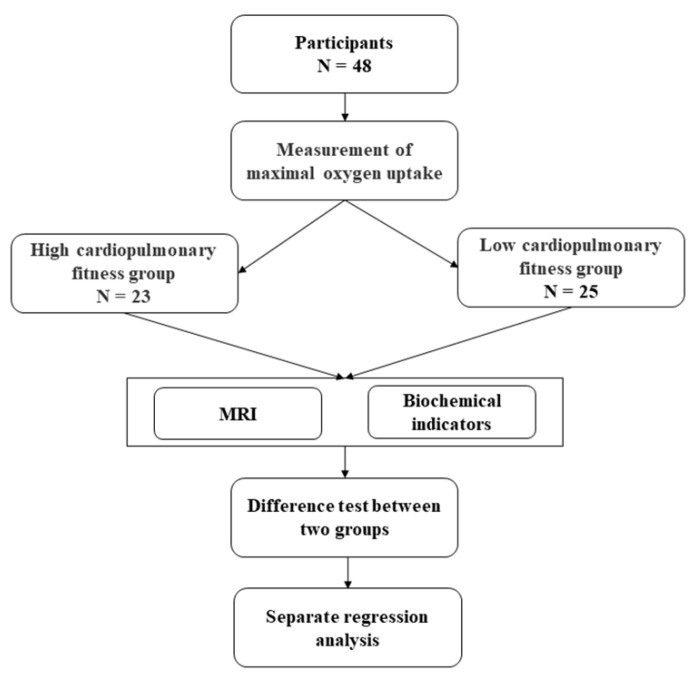
Study design.

**Figure 2 brainsci-12-00359-f002:**
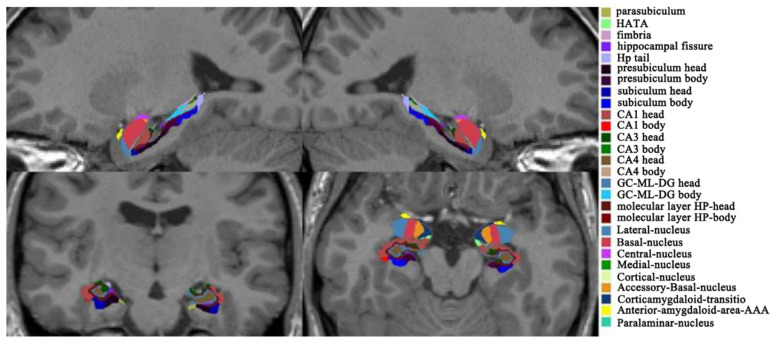
Subfields of the amygdala and hippocampus. CA, cornuammonis; HATA, hippocampal amygdala transition area; GC_ML_DG: granule cell layers of the dentate gyrus.

**Figure 3 brainsci-12-00359-f003:**
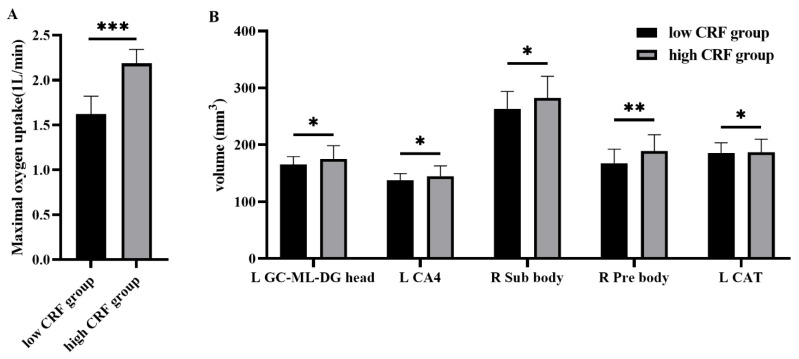
(**A**) The difference in maximal oxygen uptake between the two groups; (**B**) Differences between the hippocampal and amygdala subfields. L, left hemisphere; R, right hemisphere; CRF, cardiorespiratory fitness; GC-ML-DG head: granule cell layers of the dentate gyrus; CA4: cornuammonis 4; Sub body: subiculum body; Pre body: presubiculum body; CAT: corticoamygdaloid transition; * *p* ≤ 0.05; ** *p* ≤ 0.01; *** *p* ≤ 0.001.

**Table 1 brainsci-12-00359-t001:** Independent sample T test of demographic characteristics in the low- and high-CRF groups (mean ± SD).

	Low	High	*t*	*p*
Age	21.00 ± 1.00 (years)	20.83 ± 1.07 (years)	0.58	0.56
BMI	20.64 ± 3.34	21.65 ± 1.95	−1.26	0.20
Education	14.08 ± 0.28 (years)	14.17 ± 0.39 (years)	−0.97	0.34
Multimedia	6.04 ± 2.62 (hours)	5.57 ± 1.59 (hours)	0.75	0.46

SD: standard deviation; BMI: body-mass index; CRF: cardiopulmonary fitness; *p* < 0.05: statistical significance.

**Table 2 brainsci-12-00359-t002:** Statistical values of the independent sample T test for biochemical indexes in the low- and high-CRF groups (mean ± SD).

	Low	High	*t*	*p*
DBIL	4.9 ± 2.53 (umol/L)	6.43 ± 2.3 (umol/L)	−2.18 *	0.035
TBIL	16.56 ± 8.14 (umol/L)	21.64 ± 8.73 (umol/L)	−2.09 *	0.042
NEUT	3.19 ± 1.34 (10^9/L)	3.55 ± 1.16 (10^9/L)	−1.00	0.322
LYMPH	2.46 ± 0.56 (10^9/L)	2.56 ± 0.51 (10^9/L)	−0.64	0.528
EO	0.10 ± 0.07 (10^9/L)	0.08 ± 0.07 (10^9/L)	0.95	0.346
RBC	5.31 ± 0.53 (10^9/L)	5.87 ± 0.42 (10^9/L)	−3.97 ***	<0.001
HGB	163.20 ± 19.53 (g/L)	180.17 ± 13.41 (g/L)	−3.48 ***	<0.001
RDW-SD	42.70 ± 3.11 (%)	40.22 ± 2.56 (%)	3.00 **	0.004

TBIL: total bilirubin; DBIL: direct bilirubin; NEUT: neutrophil count; LYMPH: lymphocyte; EO: eosinophil count; RBC: red blood cells; HGB: hemoglobin; HCT: hematocrit; RDW-SD: standard deviation in red cell distribution width; * *p* ≤ 0.05; ** *p* ≤ 0.01; *** *p* ≤ 0.001.

**Table 3 brainsci-12-00359-t003:** Multiple linear regression analysis of the hippocampal subregion, amygdala subregion, and biochemical indexes within the two groups.

Dependent Variable	Predictors	B	Ser	Beta	*t*
R Sub body	Constant	152.33	44.04		3.459
	HGB	0.70	2.56	0.38	2.75
R Pre body	Constant	83.47	35.86		2.33
	HGB	0.55	0.21	0.36	2.65

HGB: hemoglobin; L: left hemisphere; R: right hemisphere; Sub body: subiculum body; Pre body: presubiculum body; *p* < 0.05, FDR correct.

**Table 4 brainsci-12-00359-t004:** Multiple linear regression analysis of the hippocampal subregion, amygdala subregion, and biochemical indexes for both groups.

	Dependent Variable	Predictors	B	Ser	Beta	*t*
Low CRF	L GC_ML_DG head	Constant	80.18	22.97		3.49
		EO	−101.75	26.48	−0.49	−3.84
	L CA4 head	Constant	65.92	19.07		3.43
		EO	−91.35	26.54	−0.53	−3.44
	R Sub body	Constant	228.79	14.37		15.92
		NEUT	10.86	4.17	0.48	2.61
	R Pre body	Constant	125.16	9.10		13.75
		NEUT	13.32	2.64	0.73	5.05
	L CAT	Constant	93.703	46.48		2.029
		EO	−744.84	302.94	−0.63	−3.35
High CRF	L GC_ML_DG head	Constant	208.01	13.17		15.80
		DBIL	−4.61	−1.74	−0.45	−2.65
	L CA4 head	Constant	60.99	41.39		1.47
		TBIL	−3.40	1.31	−0.44	−2.60

EO: eosinophil; NEUT: neutrophil; RBC: red blood cells; DBIL: direct bilirubin; TBIL: total bilirubin; L: left hemisphere; R: right hemisphere; GC-ML-DG head: granule cell layers of the dentate gyrus; CA4 head: cornuammonis 4 head; Sub body: subiculum body; Pre body: presubiculum body; CAT: corticoamygdaloid transition; *p* < 0.05, FDR corrected.

## Data Availability

Data materials can be obtained by contacting the corresponding author.

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
