# Peer review of "Changes in Hippocampus and Amygdala Volume with Hypoxic Stress Related to Cardiorespiratory Fitness under a High-Altitude Environment"

_brainsci, 2022, doi:10.3390/brainsci12030359_

Round 1
Reviewer 1 Report
The manuscript entitled “Change of hippocampus and amygdala with hypoxic stress related to cardiorespiratory fitness under high altitude environment” by Wang and coworkers deals with the interesting subject of the alterations in specific brain regions during hypoxia and the relationship with cardiorespiratory fitness.
This investigation summarizes two different spectrums of analysis, which could be causally connected, but does not complete the evaluation of either high altitude impact on RBC or neuroinflammatory mechanisms that result in alterations of brain regions involved in mood regulation.
Therefore, I suggest that title should be rephrased in order to depict the presented data (volume of hippocampus and amygdale?, neuroinflammation markers?).
Indicate units on y-axis in Fig 3.
ROIs on x-axis should replace with abbreviations that represent estimated regions.
Have you recorded mood status, life quality… in your participants in order to evaluate the significance of the clinical manifestations for the observed changes?
I suggest at least the discussing of the alterations in hippocampal and amygdale volume and its their impact on behavioral outcome, in order to emphasize the importance of this study.
Line 122 CEPT is abbreviation of cardiopulmonary exercise testing not cardiorespiratory
Have you measured any other biochemical parameters that can reveal the inflammatory profile (such as CRP, sedimentation, cytokine profile…)? If not, why?
At the same time, the authors included some unspecific parameters that do not fit with the content of this investigation (bilirubin…)?
Set the Figures and Tables in appropriate place in the Results section.
Comment all results with significance and mention p-value in the text, text must describe the data from figures and tables.
Check spelling and grammar errors.
Reviewer 2 Report
The authors have tested the cardiorespiratory fitness effect on the hippocampus and amygdala on the anatomical and cellular levels. The authors conclusively showed that high CRF groups have better results than low CRF groups. The biochemical markers also indicate that the high CRF groups have better indexes than low CRF. Their introduction and discussions are well written and cover the most critical questions. However, a few minor concerns in the study need to be addressed before acceptance.
- Hypoxia effect on neuroinflammation has already been performed with animal models in a controlled lab environment [https://link.springer.com/article/10.1007/s12035-022-02750-5]. However, patients tested here are randomly selected from Tibet, which is not a controlled environment. Hence, the patients’ epigenetics could affect the results. Authors should consider adding an explanation on this in their discussion.
- Following the last comment, were all the biochemical levels tested before and after CPET? This result could conclusively prove the effectiveness of CRFs effect on their indexes.
Some minor errors:
- Lines 106 and 107 – Should be in results. Figures 3 mentioned here is out of order.
- Line 47 -not “tress” but “stress”
- Line 71, CFR to CRF
- Line 259 – Remove “with”
There are a few more in the manuscripts. I recommend the authors to check the manuscript thoroughly for spell checks and grammatical errors.
Reviewer 3 Report
Overall I think the manuscript is interesting and may be useful to the scientific community.
I believe that the introduction deals in a correct and up-to-date way with the subject matter addressed.
As for the participants, only men are included, this is indicated as a limitation later in the discussion. I think it would be necessary to include the sample calculation used to carry out the study. If it is a convenience sample and the sample calculation was not performed, the statistical power of the participants included should be included.
Regarding the statistical analysis, it is not indicated whether the data analyzed follow a normal distribution or not, and what test was used to determine the distribution of the data.
The study indicates that multiple regression analysis was performed. In order to perform this analysis, there are some requirements that must be included: non-collinearity, parsimony, linear relationship between the numerical predictors and the response variable, normal distribution of the residuals, constant variability of the residuals (homoscedasticity), and no autocorrelation (independence). Please indicate whether these assumptions are met.
Round 2
Reviewer 1 Report
The authors sufficiently corrected the manuscript.
Reviewer 3 Report
The manuscript can be accepted in its current form.